# Blood–Brain Barrier Integrity Damage in Bacterial Meningitis: The Underlying Link, Mechanisms, and Therapeutic Targets

**DOI:** 10.3390/ijms24032852

**Published:** 2023-02-02

**Authors:** Ruicheng Yang, Jundan Wang, Fen Wang, Huipeng Zhang, Chen Tan, Huanchun Chen, Xiangru Wang

**Affiliations:** 1State Key Laboratory of Agricultural Microbiology, College of Veterinary Medicine, Huazhong Agricultural University, Wuhan 430070, China; 2Key Laboratory of Preventive Veterinary Medicine in Hubei Province, The Cooperative Innovation Center for Sustainable Pig Production, Wuhan 430070, China; 3Key Laboratory of Development of Veterinary Diagnostic Products, Ministry of Agriculture of the People’s Republic of China, Wuhan 430070, China; 4International Research Center for Animal Disease, Ministry of Science and Technology of the People’s Republic of China, Wuhan 430070, China

**Keywords:** bacterial meningitis, blood–brain barrier, brain microvascular endothelial cells, permeability, tight junction proteins

## Abstract

Despite advances in supportive care and antimicrobial treatment, bacterial meningitis remains the most serious infection of the central nervous system (CNS) that poses a serious risk to life. This clinical dilemma is largely due to our insufficient knowledge of the pathology behind this disease. By controlling the entry of molecules into the CNS microenvironment, the blood–brain barrier (BBB), a highly selective cellular monolayer that is specific to the CNS’s microvasculature, regulates communication between the CNS and the rest of the body. A defining feature of the pathogenesis of bacterial meningitis is the increase in BBB permeability. So far, several contributing factors for BBB disruption have been reported, including direct cellular damage brought on by bacterial virulence factors, as well as host-specific proteins or inflammatory pathways being activated. Recent studies have demonstrated that targeting pathological factors contributing to enhanced BBB permeability is an effective therapeutic complement to antimicrobial therapy for treating bacterial meningitis. Hence, understanding how these meningitis-causing pathogens affect the BBB permeability will provide novel perspectives for investigating bacterial meningitis’s pathogenesis, prevention, and therapies. Here, we summarized the recent research progress on meningitis-causing pathogens disrupting the barrier function of BBB. This review provides handy information on BBB disruption by meningitis-causing pathogens, and helps design future research as well as develop potential combination therapies.

## 1. Introduction

Bacterial meningitis is an inflammation of the meninges, including the dura mater, arachnoid mater, and pia mater, in response to bacterial infection [1]. It remains an important cause of the high mortality rate and incidence of neurological sequelae [2]. Case mortality rates have ranged between 5% and 40%, while between 25% and 50% of survivors have neurological conditions such as cerebral palsy, mental retardation, blindness, deafness, and seizures [3,4]. The most frequent causes of meningitis in infants and adults worldwide are *Escherichia coli* (*E. coli*), Group B Streptococcus (GBS), *Listeria monocytogenes* (*L. monocytogenes*), *Mycobacterium tuberculosis* (*M. tuberculosis*), *Streptococcus pneumoniae* (*S. pneumoniae*), *Neisseria meningitidis* (*N. meningitidis*), and *Haemophilus influenzae* type b (Hib) [5,6]. A significant new zoonotic pathogen that can also cause meningitis in humans is *Streptococcus suis* (*S. suis*) [7]. Generally, the pathological process of bacterial meningitis includes the following stages, namely mucosal colonization, microbial translocation of mucous membrane and invasion into the intravascular space, followed by intravascular survival and multiplication, reaching a high degree of bacteremia, translocation over the blood–brain barrier (BBB), and invasion of the meninges and central nervous system (CNS) [8]. In addition to the BBB, there are also a blood–cerebrospinal fluid barrier at the choroid plexus and a meningeal barrier at the subarachnoidal space between the CNS and the rest of the body. Bacteria can also enter the CNS by crossing these barriers [9]. Bacteria can then increase the BBB permeability and induce pleocytosis, which causes edema, an increase in intracranial pressure, and the release of inflammatory factors from white blood cells and other host cells that have been infiltrated [10].

The CNS is a shielded environment protected by the meninges, the vertebral column, and the skull [11]. The meninges are a membranous envelope in connective tissue whose primary function is to protect the spinal cord and brain from trauma [12]. The BBB provides a selective filter tightly regulating the exchange of water, ions, oxygen, nutrients, and other compounds between the CNS and the bloodstream [13]. In addition, it protects the CNS from invasive pathogens [14]. The basement membrane, astrocytes, microglial cells, pericytes, and the microvasculature support the brain microvascular endothelial cells (BMECs), which create the wall of the blood capillaries [15,16,17]. BMECs form many tight junctions (TJs) and some adherens junctions with adjacent cells, resulting in poorer paracellular permeability [18]. TJs, which are composed of Zonula Occludens proteins (especially ZO-1, -2, and -3), Occludins, Claudins (especially Claudin-5 and -12), and junction adhesion molecules, help to create high trans-endothelial electrical resistance (TEER) and control polarity to the BBB endothelium (Figure 1) [19,20,21]. Therefore, any deterioration or separation of these proteins from their counterparts may lead to an increase in barrier permeability [22]. In addition, astrocytes and pericytes maintain the barrier property of BMECs, but their effects on microbial entry into the CNS across the BBB appear to be limited [23].

Meningeal pathogens can penetrate the BBB and damage the BBB integrity, but such ability is lacking for non-meningitis-causing microorganisms [24]. With the availability of in vitro monolayer BMECs and in vivo mouse models, tremendous progress has been achieved up to this point in understanding the molecular interactions between meningeal infections and the BBB [25]. Meningeal pathogens have been shown to penetrate the host BBB by transcytosis (intracellular trafficking through the endothelial cells), paracytosis (through the intercellular junctional spaces), or Trojan horse mechanism (utilizing infected phagocytes as vehicles) [26]. Apart from these strategies, severe CNS inflammatory responses following microbial invasion and replication in the brain and the cytotoxicity of microbial toxins lead to the enhancement of BBB permeability, which also facilitates pathogens crossing the BBB [12]. Our limited knowledge of their etiology, particularly how pathogenic microorganisms increase BBB permeability, is a crucial factor contributing to such high mortality and morbidity. This paper provides an overview of the processes through which meningitis-causing bacteria increase BBB permeability. In order to maintain BBB integrity in CNS infectious disorders, it will be important to understand the molecular causes of increased BBB permeability. This will help create safer and more efficient therapeutic strategies.

## 2. Enhancement of BBB Permeability by Bacterial Virulence Factors

### 2.1. Bacterial Surface Structure

Lipopolysaccharide (LPS) is located on the outer leaflet of the outer membrane. It serves as the main surface component of the cell envelope of Gram-negative bacteria, responsible for activating the host’s innate immune system [27]. It was reported that LPS treatment reduced expression of TJs in BMECs and altered BBB structural integrity in vitro. Exposure to LPS decreased the expression of hydroxycarboxylic acid receptor 1 (HCAR1) and monocarboxylate transporters-1 in BMECs, and meanwhile induced interleukin (IL)-1β overproduction and a dose-dependent increase in lactate concentrations in the extracellular space, which led to the increase of BBB permeability [28]. Moreover, *E. coli* LPS affects the BBB via the crosstalk between protein kinase C (PKC) and RhoA signals, independent of the phosphatidylinositol 3-kinase (PI3K) and tyrosine kinase pathways [29]. Most recently, sulfasalazine was reported to improve the maintenance of BBB integrity and relieve *E. coli* LPS-induced inflammatory apoptosis [30]. In a rat model of Hib meningitis, intracisternal inoculation of Hib LPS (2 pg to 20 ng) resulted in dose-dependent increases in BBB permeability [31]. Outer membrane vesicles (OMV) are nano-sized spherical vesicles released by many Gram-negative bacteria, with a lipid bilayer structure that ranges in size from 20 to 250 nm. These vesicles include elements resembling those seen in bacteria, such as phospholipids, deoxyribonucleic acid, β-barrel proteins, lipoproteins, LPS, etc., because they are formed by the cell envelop of bacteria [32]. According to previous study, OMV may include a significant amount of Hib LPS, which could alter its activity. Intracisternal inoculation of Hib OMV in adult rats led to dose- and time-dependent increases in BBB permeability, much like inoculation of purified Hib LPS [33]. In addition, the capsule and peptidoglycan of Hib are also critical virulence factors to destroy host BBB integrity [34,35].

*Bacillus anthracis* (*B. anthracis*) is a sporulating Gram-positive rod that enters the human body mostly through the skin, and hemorrhagic meningitis is one of the deadly consequences of anthrax [36]. It has been recently shown that proteolytic breakdown of the monolayer hBMECs by *B. anthracis* is associated with the degradation of ZO-1. This procedure necessitates bacterial attachment to BMECs via the S-layer adhesin BslA [37].

The *S. suis* surface structure, including muramidase-released protein (MRP), factor H-binding protein (Fhb), and *S. suis* protein endopeptidase O (SsPepO), has been widely reported to destroy the integrity of BBB. MRP is a vital virulence marker of *S. suis* serotype 2 (SS2), which can bind to human fibrinogen [38]. It was discovered that the human fibrinogen-mediated adhesion and traversal ability of SS2 across BMECs are both considerably impaired by SS2 deletion of MRP. Meanwhile, measurement of the permeability to Evans blue extravasation in vivo and Lucifer yellow in vitro show that the MRP-human fibrinogen interaction dramatically enhances the BBB permeability via destroying the cell adherens junction protein p120-catenin of BMECs [39]. Another study showed that the Fhb contributed to *S. suis*-induced meningitis by interaction with globotriaosylceramide (Gb3). Gb3, also known as CD77, is restricted in expression to certain cell types, including some epithelial and endothelial cells [40]. Through Rho/Rho-associated protein kinase (ROCK) signaling, Gb3 may have an impact on the activation of myosin light chain 2 brought on by *S. suis* infection in hBMECs. Gb3 deficiency protected mice from severe brain inflammation or damage [41]. Furthermore, SsPepO also contributed to the pathogenesis of *S. suis* meningitis, which was identified as a predicted metalloendopeptidase that shares homology with the M13 peptidase family [42]. SsPepO was found to bind to human fibronectin to promote adherence and traversal ability of *S. suis* across monolayer hBMECs. Meanwhile, the SsPepO-human fibronectin-integrin interaction significantly increased the permeability of the BBB [43].

SS2 enolase was initially identified as a key glycolytic enzyme. Subsequently, enolase was identified to be expressed on the surface of *S. suis* [44]. Enolase can help pathogens infect host cells by interacting with the plasminogen receptor on the surfaces of various cell types [45]. The SS2 enolase has been shown to be important in disrupting BBB integrity by causing the release of IL-8 [46]. The extracellular adenosine of SS2 was also a contributor to promoting BBB permeability. The adenosine could activate the A1 adenosine receptor signaling pathway in BMECs and attendant cytoskeleton remodeling to damage BBB integrity. The study also found that adenosine orthologs from other bacterial species promote their translocation across BBB [47].

In addition, *Streptococcus equi* subsp. *zooepidemicus* (SEZ), which belongs to Group C *streptococcal* species, is another important animal pathogen and can cause meningitis in humans [48]. BifA is a recently identified virulence factor that facilitates SEZ adhesion to host tissue and immune evasion. BifA encodes a protein with an N-terminal RhuM domain and a C-terminal Fic domain. Fic (filamentation induced by cyclic AMP) domain-containing proteins are found in many animal and plant pathogens. It was reported that SEZ BifA’s Fic domain enables its binding and activation of cytoskeletal regulatory protein moesin. The phosphorylation of moesin could activate the downstream RhoA signaling pathway and thus destroy the integrity of BBB [49]. In summary, the surface structure of bacteria is considered the most critical factor in mediating the increase in permeability of the BBB (Figure 2).

### 2.2. Hemolysins

Hemolysin is an extracellular toxic protein produced by many Gram-negative and Gram-positive bacteria. Meningitis-causing pathogens such as *E. coli*, *L. monocytogenes*, *S. pneumoniae*, GBS, and *S. suis* can produce hemolysin [50]. As its name suggests, hemolysin is cytolytic. It binds to the host cell membrane, causing a pre-pore to form, then pierces the cell membrane and causes conformational changes within the host cell to form a mature ply pore. The mature ply pore in host cells drives protein influx and imbalances in signal transduction [51].

*E. coli* α-hemolysin is the best-studied repeat-in-toxin protein (repeat-in-toxin proteins are widespread among Gram-negative bacteria) released by the type I secretion system. It is an important virulence factor due to its cytolytic and cytotoxic activity against a diverse range of mammalian cell types (e.g., erythrocytes, monocytes, granulocytes, and endothelial cells) [52]. The α-hemolysin in meningitic *E. coli* K1 strain has been shown to reduce the TGFβ1 receptor TGFBRII and the critical transcription factor Gli2 of hedgehog signaling in BMECs, eventually leading to the BBB breakdown [53].

Another hemolysin, “suilysin”, is involved in modulating *S. suis* interactions with host cells [54]. Suilysin was discovered to cause a significant release of IL-6 and IL-8 by swine BMECs, destroying the integrity of the BBB [55]. In another study, suilysin was demonstrated to increase the paracellular permeability of BBB via the activation of group III secretory phospholipase A2 (PLA2G3) in vivo and in vitro [56].

Listeriolysin O, a pore-forming toxin generated by *L. monocytogenes*, is a particularly important virulence factor that performs many roles in guaranteeing the pathogen’s intracellular survival in hosts [57]. A result of *L. monocytogenes* infection in the CNS showed that listeriolysin O-mediated cytotoxicity against BMECs enables *L. monocytogenes* to effectively penetrate the BBB [58].

Pneumolysin is another widely studied hemolysin and a major virulence factor produced by *S. pneumoniae* [59]. When hBMECs were infected with *S. pneumoniae*, the cells rounded and detached, and the TEER of the monolayer hBMECs decreased significantly. An *S. pneumoniae* mutant deficient in pneumolysin did not affect the integrity of the hBMECs monolayer. However, purified pneumolysin-caused hBMECs monolayer damage was comparable to that caused by *S. pneumoniae* [60]. In another study, it was shown that pneumolysin causes a high expression of CREB-binding protein, which can result in the release of tumor necrosis factor-α (TNF-α) and then accelerate apoptosis of cells, which is a crucial factor contributing to BBB permeabilization [61]. In addition, pneumolysin-induced pore formation affects glial cells, altering astrocyte structure and increasing overall BBB permeability [62].

The hemolysin encoded by GBS was called β-hemolysin. It has been reported that GBS can repress the transcription of β-hemolysin under the regulation of the two-component system CovR/CovS. Moreover, the serine/threonine kinase Stk1 can phosphorylate CovR at threonine 65 to relieve the repression of β-hemolysin. Due to more β-hemolysin produced, CovR deficient GBS were more proficient in the induction of permeability and pro-inflammatory signaling pathways in BMECs [63]. In conclusion, hemolysin is one crucial virulence factor for meningitis-causing pathogens that damage the host BBB (Figure 2).

### 2.3. Enzymes

Bacteria produce a wide variety of secreted enzymes, including streptokinase, esterase, DNases, hyaluronidases, superoxide dismutase, and immunoglobulin degrading enzymes, many of which are considered virulence factors. Accumulating studies have shown that these enzymes are involved in the degradation of extracellular matrix between BMECs by pathogenic bacteria.

GBS produces a specific exotic enzyme, hyaluronidase (HylB). It was recently determined that HylB degrades hyaluronic acid into disaccharide fragments, which blocks toll-like receptor 2 (TLR2) and TLR4, preventing GBS ligands from activating the pro-inflammatory signaling pathway [64]. In GBS meningitis, it was found that the inactivation of HylB resulted in significantly decreased BBB permeability and the intravenous administration of purified HylB protein resulted in dose-dependent BBB opening [65]. In addition, research showed that type II CRISPR RNA-guided endonuclease Cas9 (Cas9) plays a vital role in GBS meningitis pathogenesis by repressing the regR regulator, further elevating HylB secretion that results in BBB integrity damage [66].

The metalloprotease immune inhibitor A (InhA), secreted by *B. anthracis*, can degrade matrix proteins. Purified InhA treatment of hBMECs resulted in a time-dependent decrease in TEER followed by ZO-1 degradation. Meanwhile, mice given purified InhA intravenously demonstrated a time-dependent Evans blue dye extravasation, leukocyte infiltration, and InhA distribution in the brain parenchyma, indicating BBB leakage and cerebral hemorrhage [22].

Eukaryotic-type serine/threonine kinases (STK) are expressed in many prokaryotes, including GBS, *M. tuberculosis*, and *S. pneumoniae*. STK was reported to regulate bacteria stress response, biofilm formation, cell wall biosynthesis, development, metabolism, antibiotic resistance, and virulence [67]. According to a recent study, the SS2 wild-type strain crossed the BBB model more easily than the STK mutant strain. STK may modulate the expression of E3 ubiquitin ligase HECTD1, increasing the degradation of Claudin-5 and allowing SS2 to cross the BBB [68]. Current studies have confirmed that bacterial enzymes can directly or indirectly affect the integrity of BBB. However, more in-depth research is still required to determine their specific contributions because of the diversity of mechanisms and functions of enzymes (Figure 2).

## 3. Host Signaling Mediators That Regulate BBB Permeability

### 3.1. Cytokines

There have been several studies on cytokines and chemokines in patients with bacterial meningitis, including the different stages of the infection process [69]. The most important factor in triggering BBB dysfunction is the formation of the CNS cytokine storm, which is caused by the excessive production of these pro-inflammatory molecules [70].

The production of various cytokines, including IL-1β, IL-6, TNF-α, and IL-8, is principally responsible for BBB breakdown during neuroinflammation [71]. TJs function alteration and BBB permeability increase are closely related to the production of cytokines (IL-1β, IL-6, and TNFα) in the brain [72]. For example, IL-6 and IL-1β were increased in the hippocampus in GBS-infected neonate rats, which increased the paracellular permeability of BMECs by decreasing TJs [73]. In addition, interferon-gamma (IFNγ), a pro-inflammatory cytokine, has been demonstrated to be a key player in the pathogenesis of experimental pneumococcal meningitis. The integrity of the BBB is impacted by IFN-modulated nitric oxide synthase 2 (NOS2), one of the many factors leading to pneumococcal meningitis pathogenesis [74]. In CNS infection with *M. tuberculosis*, it was shown that mycobacterium can induce granuloma formation on the monolayer BMECs, which led to cluster-associated destruction of the BMECs monolayer defined by mitochondrial stress, disruption of ZO-1 and Claudin-5, and enhanced transmigration of bacteria-infected cells across the BBB.

On the other hand, inhibition of TNF-α decreases the formation of clusters on BMECs and lessens damage from clusters [75]. It was found that the increase in BBB permeability induced by either meningitic *E. coli* or *S. pneumoniae* could be inhibited by anti-TNF-α antibodies [76]. Moreover, our study has shown that macrophage migration inhibitory factor (MIF) was significantly upregulated in meningitic *E. coli* infected-hBMECs. The recombinant MIF decreased the TEER values of the hBMECs monolayer dose-dependently and led to decreased expression of TJ proteins such as ZO-1 and Occludin [77].

*L. monocytogenes* cross the BBB in the form of “Trojan Horse”; therefore, macrophages’ migration and crossing the BBB is very important for *L. monocytogenes* to induce meningitis [78]. Recently, RhoA was reported to increase macrophage migration and trigger the production of IL-1β, IL-6, and TNF-α. In turn, the expression of IL-1β, IL-6, and TNF-α may facilitate the macrophage migration and adhesion across the BBB [79]. Taken together, it is clear that a variety of cytokines can increase BBB permeability by suppressing TJs, but more molecular mechanisms need to be further analyzed (Figure 3).

### 3.2. Vascularization Factors

The mammalian vascular endothelial growth factor (VEGF) family has five members: VEGF-A, -B, -C, -D, and placental growth factor (PGF). Among them, VEGF-A has gotten the most attention. VEGF-A can promote angiogenesis and neuroprotection, and induce neurogenesis [80]. It has been known to be a potent activator of vascular permeability by activating multiple signaling pathways downstream of VEGFR2 [81]. Moreover, VEGF-A has been shown to alter the expression and distribution of TJs, resulting in BBB hyperpermeability in *E. coli* meningitis [82]. Most recently, resveratrol treatment was found to maintain BBB permeability by suppressing the activation of extracellular signal-regulated protein kinases1/2 (ERK1/2)-VEGFA signaling cascade [83]. Astrocytes and pericytes seem to be the primary sources of VEGF-A in pathological conditions in CNS diseases [84]. In a *Haemophilus influenzae* type a (HiA) study, the infection activated adenosine receptors A2A and A2B in hBMECs and pericytes, causing the pericytes to release large amount of VEGF-A. The high level of VEGF-A may cause pericytes detachment and hBMECs proliferation, thereby causing BBB damage [85]. In addition, VEGF-A also participates in the breakdown of the BBB in *M. tuberculosis* meningitis, further exacerbating the disease [86,87].

Platelet-derived growth factor (PDGF)-BB possesses chemotactic, differentiating, mitogenic, and angiogenic properties and is concerned with wound healing [88]. Furthermore, it is understood that PDGF-BB regulates BBB homeostasis and is essential for preserving CNS stability [89]. However, in addition to its neuroprotective effects, studies have shown that cocaine-mediated PDGF-BB induction in hBMECs resulted in BBB damage by decreasing ZO-1 expression [90]. In our work, we were the first to document how meningitic *E. coli* caused human and mouse BMECs to experience a time-dependent elevation of PDGF-BB, which led to TJ disarrangement [91].

Angiopoietin-like protein-4 (Angptl4) is a secreted glycoprotein with a physiological role in lipid metabolism. Angptl4 was reported to involve vascular permeability, angiogenesis, and inflammatory responses in different tissues [92]. In the field of cancer research, there are contradictory reports about the relevance of Angptl4 in regulating vascular permeability [93]. Our recent study has demonstrated that Angptl4 was markedly elevated in meningitic *E. coli* infection of hBMECs as well as in a mice model, and the induction of Angptl4 contributes to BBB disruption via ARHGAP5/RhoA/MYL5 signaling cascade [94]. Understanding the characterization of these vascularization targets involved in CNS infectious diseases, such as VEGF-A, PDGF-BB, and ANGPTL4 will open new opportunities for using these as potential therapeutic targets for bacterial meningitis (Figure 3).

### 3.3. Apoptosis Related Factors

A key cellular response called apoptosis is crucial for development and stability. Genetic studies have shown that the loss of pro-apoptotic genes leads to abnormalities in the CNS. In addition, the removal of dysfunctional cells is vital to the stability of the internal tissues and organs environment. However, when improperly controlled, apoptosis can potentially advance or even exacerbate disease processes [95]. By activating the extrinsic route, pro-inflammatory cytokines such as TNF-α, CD40/CD40 ligand, CD47, and its ligand thrombospondin-1 cause the apoptosis of BMECs [96]. Meningitis-causing pathogens, including *S. pneumoniae*, *Haemophilus parasuis* (*H. parasuis*), and *S. suis*, can induce BMECs apoptosis via several mechanisms. For instance, two different mitochondrial pathways are activated by *S. pneumoniae* to initiate apoptosis: a caspase-3-dependent pathway that is triggered by the physical contact between the bacteria and the BMECs, and a caspase-3-independent pathway that is triggered by pro-inflammatory components of the bacterial cell wall, or by the release of toxins like pneumolysin and H_2_O_2_ [97]. BMECs apoptosis was also detected during *H. parasuis* infection. In a time- and dose-dependent manner, *H. parasuis* induces caspase-3-mediated apoptosis of porcine BMECs [98].

It has been determined that the virulence factor SS2 Enolase affects BBB integrity. According to a recent study, SS2 Enolase binds to the 40S ribosomal protein SA on the surface of porcine BMECs. This causes the intracellular p38/ERK-eIF4E signaling pathway to be activated, which encourages the expression of the heat-shock protein family D member 1 (HSPD1) and starts the apoptosis process in the BMECs. This increases the BBB permeability and, in turn, facilitates bacterial invasion [99].

In addition, the type VI secretion system (T6SS) has recently been identified and characterized in several Gram-negative pathogens. It represents a complex secretion machinery that contributes to pathogenicity in many bacteria [100]. The Hcp1 was secreted in a T6SS-dependent manner in meningitic *E. coli* [101]. It was reported that Hcp1 could induce cytokines release, cytoskeleton rearrangement, and apoptosis in BMECs to damage BBB integrity [102]. BMECs apoptosis is a complex process involving distinct intracellular signaling pathways. At present, there are few studies on the apoptosis of BMECs in the field of bacterial meningitis, and further investigation is still needed (Figure 3).

### 3.4. Transcription Factors

HIF-1, a transcriptional factor, is linked to a variety of cerebral vascular pathological disorders. HIF-1 is a heterodimeric complex, which is mainly comprised of O_2_-sensitive α subunits (HIF-1α) and shared β subunits (HIF-1β), the latter being constitutively expressed under hypoxia [103]. HIF-1 could regulate transcriptional activation of several genes responsive to the lack of oxygen, such as glucose transporters, VEGF, glycolytic enzymes, and erythropoietin [104]. One of the most popular HIF-1 target genes in vascular biology is VEGF. It was reported that the upregulation of HIF-1α/VEGF pathway during *S. pneumoniae* infection is associated with BBB opening. In *S. pneumoniae*-infected mice, therapeutic rescue with the HIF-1 inhibitor echinomycin increased survival and enhanced BBB performance [105].

A zinc-finger transcription inhibitor known as Snail family transcriptional repressor 1 (SNAI1) is involved in a wide range of physiological and pathological processes, such as healthy embryonic development, epithelial injury healing, and cancer spread [106]. A growing number of studies have confirmed the role of SNAI1 in cell junctions. Overexpression of SNAI1 has been shown to destroy the top complex of vascular endothelial cells [107]. Moreover, SNAI1 can negatively regulate the expression of Claudin-5 and Occludin [108,109]. Most recently, GBS-infected hBMECs were found to increase the expression of SNAI1 that mediated the degradation of ZO-1, Occludin, and Claudin-5, and disrupted endothelial barrier integrity in cultured hBMECs [110]. In our investigation, we similarly showed that meningitic *E. coli* induces SNAI1 and that SNAI1 negatively regulates the junctional proteins ZO-1, Occludin, Claudin-5, and β-catenin. Although Snail-1 knocking-down did not fully restore the decreased expression of TJs, this reflected a negative effect of Snail-1 on the TJs to a certain degree [82]. The studies mentioned above show that transcription factors are important endogenous regulators in charge of controlling BBB permeability. However, the precise regulatory mechanisms involving other essential transcription factors in BBB damage still require more experimental endeavors (Figure 3).

### 3.5. Metalloproteinases

A class of zinc-dependent endopeptidases known as matrix metalloproteinases (MMPs) has been found to be an important regulator of the BBB integrity during bacterial meningitis [111]. Studies on CNS infectious diseases have concentrated on MMP-8 (collagenase-2) and MMP-9 (gelatinase B), as they both have the capacity to degrade basement membranes due to their substrate specificity for type IV collagen, laminin, and fibronectin, the main components of basal lamina surrounding cerebral vessels [112]. In vitro and in vivo, MMP-9 can break down Claudin-5, Occludin, and ZO-1, contributing to the breakdown of TJs [113]. Experimental evidence suggests that MMP-9, which is implicated in the breakdown of the BBB, is primarily produced by BMECs and infiltrating neutrophils [114]. For example, *M. tuberculosis* causes the breakdown of type IV collagen and decreases expression of TJs to increase the BBB permeability [115], which is driven by *M. tuberculosis*-dependent secretion of MMP-9 [116]. Except for causing direct degradation of the basement membrane and TJs, MMP-9 affects expression of TJs by inhibiting the Sonic hedgehog (Shh) signaling pathway in BMECs as well [117]. In addition, MMP-9 was found to contribute to brain damage associated with *N. meningitides* meningitis significantly, and inhibition of MMP-9 reduced intracranial complications in mice suffering from *N. meningitides* meningitis [118]. In another study, the infection of BMECs with *N. meningitides* could enhance permeability, which was accompanied by an increase in MMP-8 activity in supernatants taken from infected cells. MMP-8 was involved in the proteolytic cleavage of the TJs Occludin, causing it to vanish from the cell periphery. Moreover, MMP-8 affected cell adherence to the underlying matrix [119]. In a study of *S. suis* meningitis, dexamethasone was reported to significantly prevent *S. suis*-induced protein and morphological TJs alterations via attenuating MMP-3 expression [120].

In addition, a disintegrin and metalloprotease with thrombospondin type I repeats-13 (ADAMTS13) was reported to cleave a large polymeric adhesion protein von Willebrand factor, which was synthesized in vascular endothelial cells, maintaining the CNS function [121]. In *L. monocytogenes*-infected lambs, significantly elevated levels of ADAMTS-13 may help to control and safeguard BBB integrity and CNS cells from listeric encephalitis. Furthermore, increased ADAMTS-13 expression may be essential for promoting the survival of glia and neurons [122]. Together, these studies established that metalloproteinase activity is crucial in disassembling or maintaining cell junction components during meningitis-causing bacteria infection (Figure 3).

### 3.6. Non-Coding RNA

It was previously thought that only proteins were responsible for controlling BBB permeability, but non-coding RNA (ncRNA) has lately come to light as a crucial regulatory component of this process [123,124]. Many species of ncRNAs, including microRNAs (miRNAs), long ncRNAs (lncRNAs), and circular ncRNAs (circRNAs), can directly or indirectly influence BBB integrity, which may hold therapeutic potential nucleic acid targets in the context of bacterial meningitis [15]. For example, our team discovered that circ_2858 was upregulated in BMECs after exposure to meningitic *E. coli* and showed that this circ_2858 might promote VEGFA production by actively competing with miR-93-5p, disrupting the TJs and impairing the BBB [125]. In another work, we revealed that meningitic *E. coli* infection-induced lncRSPH9-4 exacerbated TJ disruption in BMECs, most likely via the miR-17-5p/MMP3 axis [126]. Although miRNAs have been extensively studied as potential therapeutic targets, at the time of writing, new regulatory ncRNAs such as lncRNAs and circRNAs have received more attention and are still in the early phases of research. A few investigations on the modulation of BBB permeability by lncRNAs and circRNAs described indirect pathways involving miRNAs. The lncRNAs and circRNAs provide stable molecular targets that act as the magnet for the miRNAs, impairing the regulatory functions of the specific miRNAs to which the lncRNAs or circRNAs bind [17]. As a result, more strategies for protecting BBB permeability through lncRNAs and circRNAs need to be investigated (Figure 3).

### 3.7. Pattern-Recognition Receptors

Pattern-recognition receptors are believed to induce the expression of inflammatory factors initiating the brain immune injury. One of the most significant immune defense lines against infectious illnesses, TLRs are crucial for host defense [127]. TLR2 is involved in cell activation by the cell wall and membrane components of Gram-positive bacteria, such as lipoproteins, lipoteichoic acid, and peptidoglycan [128]. In the mouse *S. pneumoniae* meningitis model, the bacterial infection caused the alteration of BBB permeability in both wild-type and TLR2 deficient mice and the higher Evans blue concentration in the brains of TLR2 deficient mice, compared with the control mice. This indicates that the activation of TLR2 helps increase the permeability of the BBB [129].

The nucleotide-binding and oligomerization domain 2 protein (NOD2) belongs to the NOD-like receptor (NLR) family [130]. Studies have found that *S. pneumoniae* enhancement of the BBB permeability is closely related to the upregulated expression of NOD2 [131]. Since pattern-recognition receptors play crucial roles in stimulating the secretion of pro-inflammatory cytokines and subsequently the development of pro-inflammatory responses, inhibiting the activation of such pattern-recognition receptors, like TLR2 and NOD2, can effectively reduce the neuroinflammatory response and maintain the stability of BBB (Figure 3).

### 3.8. Others

Reactive oxygen species (ROS) are chemically reactive molecules containing oxygen that are typical byproducts of aerobic metabolism [132]. ROS are produced by immune cells and are essential for innate host defense as effectors due to their toxic action against pathogens. However, unmanaged ROS regulation during infection may result in persistent inflammatory conditions and diseases [133]. *Staphylococcus aureus* (*S. aureus*) is a common opportunistic pathogen that can cause CNS infection and elevated paracellular permeability. *S. aureus* infection of hBMECs resulted in the dose-dependent release of cytokines/chemokines (TNF-α, IL-6, macrophage cationic peptide 1 (MCP-1), C-X-C motif chemokine ligand 10 (CXCL10), and thrombomodulin), as well as the reduction of expression of TJs (Claudin-5 and ZO-1). These events were linked to the induction of ROS within hBMECs by *S. aureus* [134].

A heme-containing peroxidase known as myeloperoxidase (MPO) is largely expressed in neutrophils and to a lesser extent in monocytes. In several inflammatory illnesses, MPO has been shown to act as a local mediator of tissue injury and the inflammation, and plays a vital role in microbial killing by neutrophils [135]. Patients with bacterial meningitis showed elevated systemic and local MPO, which was accentuated during the feverish episodes. Reacting with cell-matrix metalloproteinase, MPO may contribute to BBB dysfunction [136].

Members of the tripartite motif (TRIM) protein family have a role in a number of biological functions, such as apoptosis, oncogenesis, development, differentiation, and cell proliferation [137]. Because of its wide-ranging role in triggering innate immune responses, TRIM32, a member of the TRIM protein family, is a potential candidate for causing broad and unbalanced cytokine production [138]. Following *S. suis* infection, TRIM32 deficiency markedly decreased bacteremia and the production of pro-inflammatory cytokines, shielding the infected mice from the streptococcal toxic shock-like syndrome. Additionally, it was discovered that during the early stages of *S. suis* infection, TRIM32 loss increased the BBB’s permeability and the recruitment of inflammatory monocytes [139]. This indicates that TRIM32 both positively and negatively regulates *S. suis* meningitis. Therefore, the function of TRIM32 in *S. suis* infection still needs further study to answer (Figure 3).

## 4. Conclusions

In the past few decades, the treatment of bacterial meningitis has mainly focused on killing invasive bacteria by using antibiotics and reducing the CNS inflammatory response by using corticosteroids [140]. However, the mortality and morbidity due to bacterial meningitis remains unacceptably high. Importantly, because of the immature development of the host defense system and BBB in neonates, the complications caused by both factors released by multiplying bacteria and as a result of the inflammatory host response to bacterial components are severe [141]. Because of the traditional treatment’s shortcomings in terms of efficacy, some new treatment strategies for bacterial meningitis have been proposed in recent years. A multimodal treatment concept that targets different steps of the pathophysiologic cascade, such as using non-bacteriolytic but bactericidal antibiotics (e.g., rifampicin and daptomycin) to limit bacterial component release, decreasing neutrophil life-span to reduce leukocyte accumulation, blocking a central proinflammatory factor (e.g., IL-1β, TNF-α, and MCP-1), and maintaining BBB permeability (e.g., vascularization factors and MMPs), represents a promising approach in the successful bacterial meningitis treatment [142]. Despite the existence of above novel treatment strategies, future work on novel therapeutic targets is still required.

In order to maintain brain homeostasis under varied circumstances, BBB permeability is highly dynamic and responsive to many internal and external cues. Changes in BBB permeability are typically caused by blood-borne substances, such as bacterial metabolites, hormones, or cytokines, which either directly affect the brain endothelium or induce an inflammatory response that leads to BMECs dysfunction [143]. Damage to the integrity of the BBB accelerates the CNS infectious diseases because this dysfunction can promote the infiltration of the leukocyte and pathogens into the CNS and accelerate the development of the disease. Therefore, the pathological analysis of BBB disruption and exploring the potential molecular targets to maintain BBB permeability are indispensable and may be a potential strategy for managing bacterial meningitis. The emphasis on possible BBB alterations in pathological and pathogenic scenarios could help design novel therapeutic strategies and optimize clinical drug administration practices.

## Figures and Tables

**Figure 1 ijms-24-02852-f001:**
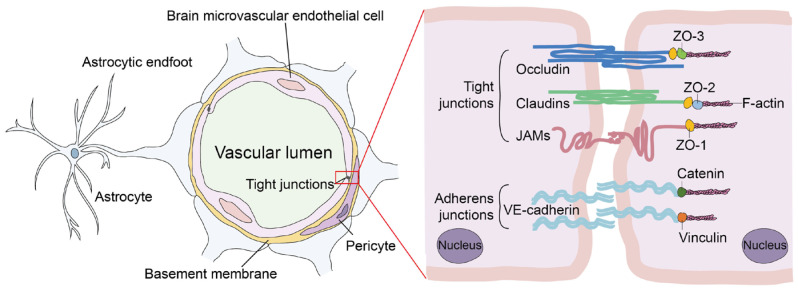
Schematic architecture of cellular components and molecules involved in the regulation of BBB integrity.

**Figure 2 ijms-24-02852-f002:**
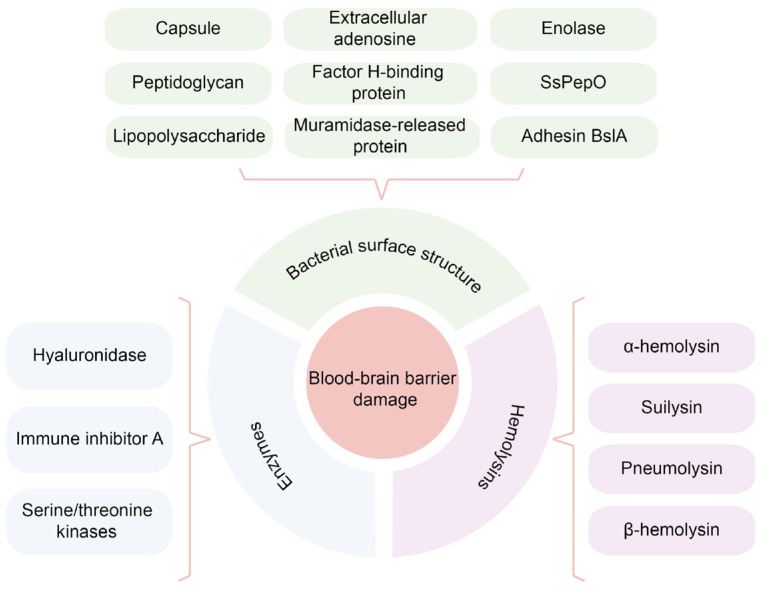
A brief summary of bacterial virulence factors that affecting the BBB permeability.

**Figure 3 ijms-24-02852-f003:**
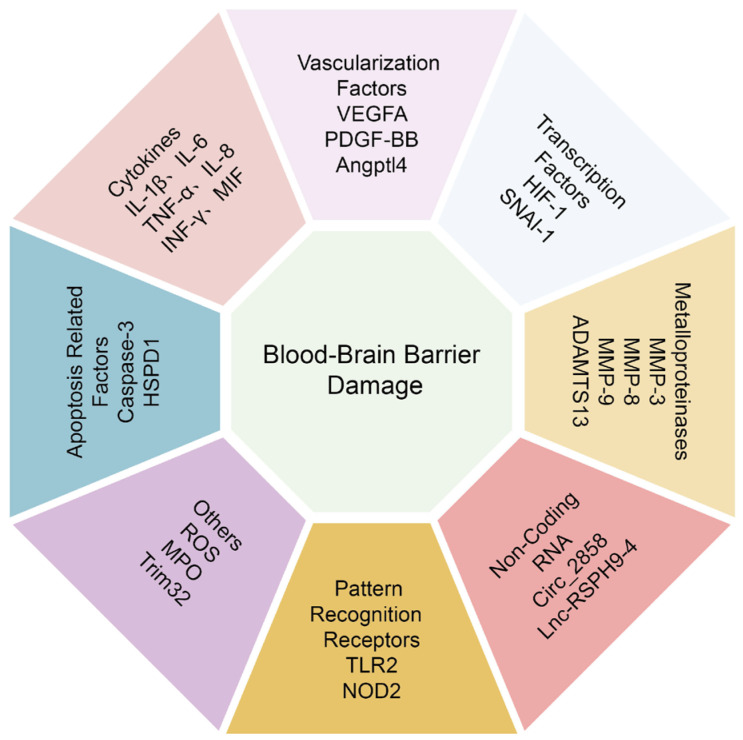
A brief summary of host signaling mediators that regulating the BBB permeability.

## Data Availability

Not applicable.

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
