# Peer review of "Blood–Brain Barrier Integrity Damage in Bacterial Meningitis: The Underlying Link, Mechanisms, and Therapeutic Targets"

_ijms, 2023, doi:10.3390/ijms24032852_

Round 1
Reviewer 1 Report
Dear authors,
The manuscript titled "Blood-brain barrier integrity damage in bacterial meningitis: the underlying link, mechanisms, and therapeutic targets," presents a compendium of comprehensive information, with a logical and orderly sequence in the presentation of information. Very isolated typographical errors are observed, such as writing E.coli in italics. They do not detract from the relevance of the document. This reviewer considers that the document lacks appropriate schemes to explain and present the information in a summarized way, providing an overview of the different approaches regarding the damage to the integrity of the blood-brain barrier presented.
The work can be accepted for publication, after a significant improvement of the images presented and a minor correction of the typography of the text.
Author Response
The manuscript titled "Blood-brain barrier integrity damage in bacterial meningitis: the underlying link, mechanisms, and therapeutic targets," presents a compendium of comprehensive information, with a logical and orderly sequence in the presentation of information. Very isolated typographical errors are observed, such as writing E. coli in italics. They do not detract from the relevance of the document. This reviewer considers that the document lacks appropriate schemes to explain and present the information in a summarized way, providing an overview of the different approaches regarding the damage to the integrity of the blood-brain barrier presented. The work can be accepted for publication, after a significant improvement of the images presented and a minor correction of the typography of the text.
Re:
Thanks for your careful review. We have improved the images and modified the typography of the text. Please check the revised manuscript.
Reviewer 2 Report
The Study entitled " Blood-bran-barrier integrity damage in bacterial meningitis: the underlying link, mechanism and therapeutic targets
This review present mainly two factors which affect the BBB permeability i.e bacterial factors and host signaling mediators.
The study looks narrative review rather than systematic review.
The mages just provides the names of bacterial and host facts.
As mentioned in the title, therapeutic targets are not clearly mentioned.
The work is superficial collection lacking systematic presentation using appropriate tables.
Hence I recommend rejection of this work.
Author Response
The Study entitled "Blood-bran-barrier integrity damage in bacterial meningitis: the underlying link, mechanism and therapeutic targets". This review present mainly two factors which affect the BBB permeability i.e bacterial factors and host signaling mediators. The study looks narrative review rather than systematic review. The mages just provides the names of bacterial and host facts. As mentioned in the title, therapeutic targets are not clearly mentioned. The work is superficial collection lacking systematic presentation using appropriate tables. Hence I recommend rejection of this work.
Re:
Thanks for your comments and helpful suggestions.
In this review, we have indeed collected a large number of research data on blood-brain barrier disruption in different pathogens and diseases, and summarized the common factors that cause changes in blood-brain barrier permeability. We believe that our work has a certain significance for scholars engaged in the field of blood-brain barrier permeability research. At least to a certain extent, our work has systematically combed the pathogenic and host factors involved in barrier dysfunction, which will help to understand the causes and associations behind different central nervous system disorders more comprehensively, and promote the development of better targeted prevention and treatment strategies. At the same time, we also believe that these reviews and generalizations are necessary for young scholars who are just engaged in this research field.
Of course, as you mentioned, there is no clear mention in terms of therapeutic targets. We have sorted out and improved this part. We have supplemented include a section covering current and potential future therapeutic targets and strategies in the review. Please check the revised manuscript.
Reviewer 3 Report
Dear Authors,
this is a well-written manuscript, providing a comprehensive review of the pathophysiologic mechanisms involved in the blood-brain barrier (BBB) integrity damage in bacterial meningitis. This interesting data may become an impulse for future therapeutic targets with practical implementation in the clinical setting. I have no questions or queries pertaining to your manuscript.
Best Regards
Author Response
This is a well-written manuscript, providing a comprehensive review of the pathophysiologic mechanisms involved in the blood-brain barrier (BBB) integrity damage in bacterial meningitis. This interesting data may become an impulse for future therapeutic targets with practical implementation in the clinical setting. I have no questions or queries pertaining to your manuscript.
Re:
Thank you for the kind review and positive comments on our manuscript.
Reviewer 4 Report
This is a very interesting and comprehensive review, looking at the possible mechanisms involved in the damage of the BBB integrity in bacterial meningitis.
Some comments that would need to be looked in the introduction:
Line 38 - Meninges – dura mater, arachnoid mater, and pia mater.
Lines 40 and 41 – mortality refers to neonatal cases? Or overall cases? However, adding information for neonatal cases of meningitis seems a bit out of context.
Line 55 – use the correct word – vertebral column instead of spine
Line 57 – conceptual error – BBB is not part of the meninges
To be considered for discussion:
- Although still controversial - if the BBB characteristics differ in the adult and developing brain, that could account for a difference in the integrity of the BBB and outcomes between neonatal cases and adult cases.
- Possibility of circumventricular organs as an additional potential site for bacterial entry
Author Response
This is a very interesting and comprehensive review, looking at the possible mechanisms involved in the damage of the BBB integrity in bacterial meningitis. Some comments that would need to be looked in the introduction:
Line 38 - Meninges - dura mater, arachnoid mater, and pia mater.
Re:
Thanks for your correction. Please check the revised manuscript Line 38.
Lines 40 and 41 - mortality refers to neonatal cases? Or overall cases? However, adding information for neonatal cases of meningitis seems a bit out of context.
Re:
Sorry for our negligence, and we have modified this information here according to the reference. Mortality refers to overall cases. Please check the revised manuscript Line 40-42.
Line 55 - use the correct word - vertebral column instead of spine.
Re:
Thanks. We have corrected it. Please check the revised manuscript Line 60.
Line 57 - conceptual error - BBB is not part of the meninges
Re:
Thanks for your kind correction. We have modified this expression. Please check the revised manuscript Line 62.
To be considered for discussion:
-Although still controversial - if the BBB characteristics differ in the adult and developing brain, that could account for a difference in the integrity of the BBB and outcomes between neonatal cases and adult cases.
Re:
Thanks for your constructive suggestions. Yes, in our opinion, because of the immature development of the host defense system and BBB in neonates, the complications caused by both factors released by multiplying bacteria and as a result of the inflammatory host response to bacterial components are severe. As per your suggestions, we have supplemented this discussion in the latest manuscript, please check Line 481-484.
-Possibility of circumventricular organs as an additional potential site for bacterial entry.
Re:
Thanks for your question. In some disease models, bacteria can enter into the circumventricular organs. However, since our review primarily focuses on the BBB integrity damage, we thus do not discuss too much on the bacterial entry here in the manuscript. But thanks for your suggestive comments.
Reviewer 5 Report
The review describes recent literature about blood brain barrier functions during infections with meningitis-causing pathogens.
It would be interesting to describe in more detail whether any evidence is available about the early events in bacterial meningitis: what are the factors that allow initial pathogen spread into the meningeal space?
Author Response
The review describes recent literature about blood brain barrier functions during infections with meningitis-causing pathogens. It would be interesting to describe in more detail whether any evidence is available about the early events in bacterial meningitis: what are the factors that allow initial pathogen spread into the meningeal space?
Re:
Thanks for your suggestion. Yes, that would be a quite systemic processes in the development of bacterial meningitis. Before bacterial spreading into the meningeal space, the early events in bacterial meningitis includes mucosal colonization, microbial translocation of mucous membrane and invasion into the intravascular space, followed by intravascular survival and multiplication, and reached a high degree of bacteremia. We just have supplemented this description in the latest manuscript, please check Line 50-52.
Reviewer 6 Report
In this review the authors provide a thorough compendium of the different factors contributing to decrease of blood-brain barrier integrity during bacterial meningitis. Among these factors they include both bacterial virulence factors and host signaling mediators. I have the following comments that should be considered by the authors:
One topic of the review is “therapeutic targets”. In the Abstract the authors write that “targeting pathological factors contributing to enhanced BBB permeability is a beneficial therapeutic adjunct to antimicrobial therapy in improving the outcome of bacterial meningitis.” This interesting aspect of BBB damage during bacterial meningitis should get more attention in the review. I suggest that the authors include a section covering current and potential future therapeutic targets and strategies in the review before the “Conclusion” section.
Although the review focuses on the BBB, the authors should point out at least in the introduction that more than one barrier between the CNS and the rest of the body exist (which include, besides the blood-brain barrier at the brain microvasculature, among others a blood–cerebrospinal fluid barrier at the choroid plexus and a meningeal barrier at the subarachnoidal space; reviewed in Saunders et al. Physiology and molecular biology of barrier mechanisms in the fetal and neonatal brain. J. Physiol. 2018, 596, 5723–5756). E.g. the authors write in lines 48-51 that “generally, the pathological process of bacterial meningitis includes … translocation through the blood-brain barrier”, but CNS entry by bacteria can also be achieved by crossing of one of the other barriers.
In lines 57-58 the authors write that “the BBB is part of the meninges”. I believe this is anatomically not correct.
In lines 63-65 the authors write that “BMECs form many tight junctions (TJs) and some adherent junctions with adjacent cells, resulting in poorer paracellular permeability and lower pinocytosis/transcytosis ability [16]”. Is it correct that the tight junctions and adherens junctions are responsible for lower pinocytosis/transcytosis ability? Also, would Ref. [16] be a good reference here? I cannot find any reference to pinocytosis/transcytosis in [16].
It is mentioned in lines 65-66 that the TJs are composed of Claudins, especially Claudin-3, -5, and -12. Importantly, brain endothelial cells do not express claudin-3 mRNA and detection of claudin-3 protein at the BBB is rather due to junctional reactivity of anti-claudin-3 antibodies to an unknown antigen still detected in claudin-3-/- brain endothelium (Castro Dias et al. Claudin-3-deficient C57BL/6J mice display intact brain barriers. Sci Rep. 2019 Jan 18;9(1):203).
In a few cases I would suggest to add some information. Also, sometimes proteins/factors are mentions without an explanation of their function. Line 107: “In Hib meningitis, intracisternal inoculation of Hib LPS…”; please add which animal model was used. Lines 383-385: “For example, M. tuberculosis causes…”; please add the MMP(s) involved and their source. Line454-455: “reduced expression of Js (VE-Cadherin, Claudin-5, and ZO-1)”; VE-Cadherin is a component of adherens junctions. Also, please give a short description for GPR81 (line 100), SsPepO (line 125), ply pore (line 169), repeat in toxin (Line 171), Cas9 (line 222).
Typos / corrections:
Line 63: adherent junctions > adherens junctions
Line 65: Zonula Occludens protein-1 > Zonula Occludens proteins
Lines 65-69: There seems to be something grammatically incorrect with this sentence.
Line 72: entry into the BBB > entry into the CNS across the BBB
Line 77: invade the BBB > penetrate the BBB
Line 78: non-meningitis microorganisms > non-meningitis-causing microorganisms
Line 80: in vitro, in vivo > should be written in italics
Line 89: increased > increase
Line 105: (PI3K)and > (PI3K) and
Line 106: E. coli > should be written in italics
Lines 117-119: I would suggest to add the sentence “In addition, the capsule and peptidoglycan of Hib…” to the end of the the paragraph before this sentence, which is dealing with Hib as well, and not with B. anthracis.
Line 129: in vivo > should be written in italics
Line 142: I would start a new paragraph with the sentence “SS2 enolase was initially…”
Line 143: to express on > to be expressed on
Line 154: Bifa was > Bifa is
Line 165: Meningitis pathogens > Meningitis-causing pathogens
Line 178: has been involved > is involved
Line 194: CERB-binding protein > CREB-binding protein
Line 267: TJs such as ZO-1 and Occludin > TJ proteins such as ZO-1 and Occludin
Line 269: L. monocytogenes > should be written in italics
Line 276: Vascularization Factor > Vascularization Factors
Lines 284-285: extracellular regulated > extracellular signal-regulated
Line 287: In the Haemophilus influenzae > In a Haemophilus influenzae
Line 291: M. tuberculosis > should be written in italics
Line 306: as well as in mice model > as well as in a mice model
Line 320: Meningitis pathogens > Meningitis-causing pathogens
Line 320: H. parasuis > should be written in italics
Line 336: It represents complex secretion machinery > It represents a complex secretion machinery
Line 346 Transcription Factor > Transcription Factors
Line 356: S. pneumonia > should be written in italics
Line 375: Metalloproteinase > Metalloproteinases
Line 395: In the study of S. suis meningitis > In a study of S. suis meningitis
Line 401: L. Monocytogenes > L. monocytogenes
Line 427 Pattern-Recognition Receptor > Pattern-Recognition Receptors
Line 428 “Pattern-recognition receptors are believed to be expressed and may produce… > I’m not sure if this sentence transports the intended meaning.
Line 468: In the study of S. suis > In a study of S. suis
Author Response
In this review the authors provide a thorough compendium of the different factors contributing to decrease of blood-brain barrier integrity during bacterial meningitis. Among these factors they include both bacterial virulence factors and host signaling mediators. I have the following comments that should be considered by the authors:
One topic of the review is “therapeutic targets”. In the Abstract the authors write that “targeting pathological factors contributing to enhanced BBB permeability is a beneficial therapeutic adjunct to antimicrobial therapy in improving the outcome of bacterial meningitis.” This interesting aspect of BBB damage during bacterial meningitis should get more attention in the review. I suggest that the authors include a section covering current and potential future therapeutic targets and strategies in the review before the “Conclusion” section.
Re:
Thanks for your suggestion. We have added a part of section covering current and potential future therapeutic targets and strategies in our latest manuscript. Please check the revised Line 478-493.
Although the review focuses on the BBB, the authors should point out at least in the introduction that more than one barrier between the CNS and the rest of the body exist (which include, besides the blood-brain barrier at the brain microvasculature, among others a blood–cerebrospinal fluid barrier at the choroid plexus and a meningeal barrier at the subarachnoidal space; reviewed in Saunders et al. Physiology and molecular biology of barrier mechanisms in the fetal and neonatal brain. J. Physiol. 2018, 596, 5723–5756). E.g. the authors write in lines 48-51 that “generally, the pathological process of bacterial meningitis includes … translocation through the blood-brain barrier”, but CNS entry by bacteria can also be achieved by crossing of one of the other barriers.
Re:
Thanks for your constructive advice. We have modified the above background, please check the revised Line 54-56.
In lines 57-58 the authors write that “the BBB is part of the meninges”. I believe this is anatomically not correct.
Re:
Thanks for your kind correction. We have modified this expression. Please check the revised manuscript Line 62.
In lines 63-65 the authors write that “BMECs form many tight junctions (TJs) and some adherent junctions with adjacent cells, resulting in poorer paracellular permeability and lower pinocytosis/transcytosis ability [16]”. Is it correct that the tight junctions and adherens junctions are responsible for lower pinocytosis/transcytosis ability? Also, would Ref. [16] be a good reference here? I cannot find any reference to pinocytosis/transcytosis in [16].
Re:
Thanks for your kind question. We have rechecked and revised the description here and updated the appropriate reference. Please check the revised manuscript Line 67-69.
It is mentioned in lines 65-66 that the TJs are composed of Claudins, especially Claudin-3, -5, and -12. Importantly, brain endothelial cells do not express claudin-3 mRNA and detection of claudin-3 protein at the BBB is rather due to junctional reactivity of anti-claudin-3 antibodies to an unknown antigen still detected in claudin-3-/- brain endothelium (Castro Dias et al. Claudin-3-deficient C57BL/6J mice display intact brain barriers. Sci Rep. 2019 Jan 18;9(1):203).
Re:
Thanks for your professional suggestion. We have carefully read this reference and made some modifications, and added this reference to the latest manuscript. Please check the revised manuscript Line 70.
In a few cases I would suggest to add some information. Also, sometimes proteins/factors are mentions without an explanation of their function. Line 107: “In Hib meningitis, intracisternal inoculation of Hib LPS…”; please add which animal model was used. Lines 383-385: “For example, M. tuberculosis causes…”; please add the MMP(s) involved and their source. Line454-455: “reduced expression of Js (VE-Cadherin, Claudin-5, and ZO-1)”; VE-Cadherin is a component of adherens junctions. Also, please give a short description for GPR81 (line 100), SsPepO (line 125), ply pore (line 169), repeat in toxin (Line 171), Cas9 (line 222).
Re:
Thanks for your advice. We have added the above information in our revised manuscript, please check the latest version.
Typos / corrections:
Line 63: adherent junctions > adherens junctions
Line 65: Zonula Occludens protein-1 > Zonula Occludens proteins
Lines 65-69: There seems to be something grammatically incorrect with this sentence.
Line 72: entry into the BBB > entry into the CNS across the BBB
Line 77: invade the BBB > penetrate the BBB
Line 78: non-meningitis microorganisms > non-meningitis-causing microorganisms
Line 80: in vitro, in vivo > should be written in italics
Line 89: increased > increase
Line 105: (PI3K)and > (PI3K) and
Line 106: E. coli > should be written in italics
Lines 117-119: I would suggest to add the sentence “In addition, the capsule and peptidoglycan of Hib…” to the end of the the paragraph before this sentence, which is dealing with Hib as well, and not with B. anthracis.
Line 129: in vivo > should be written in italics
Line 142: I would start a new paragraph with the sentence “SS2 enolase was initially…”
Line 143: to express on > to be expressed on
Line 154: Bifa was > Bifa is
Line 165: Meningitis pathogens > Meningitis-causing pathogens
Line 178: has been involved > is involved
Line 194: CERB-binding protein > CREB-binding protein
Line 267: TJs such as ZO-1 and Occludin > TJ proteins such as ZO-1 and Occludin
Line 269: L. monocytogenes > should be written in italics
Line 276: Vascularization Factor > Vascularization Factors
Lines 284-285: extracellular regulated > extracellular signal-regulated
Line 287: In the Haemophilus influenzae > In a Haemophilus influenzae
Line 291: M. tuberculosis > should be written in italics
Line 306: as well as in mice model > as well as in a mice model
Line 320: Meningitis pathogens > Meningitis-causing pathogens
Line 320: H. parasuis > should be written in italics
Line 336: It represents complex secretion machinery > It represents a complex secretion machinery
Line 346 Transcription Factor > Transcription Factors
Line 356: S. pneumonia > should be written in italics
Line 375: Metalloproteinase > Metalloproteinases
Line 395: In the study of S. suis meningitis > In a study of S. suis meningitis
Line 401: L. Monocytogenes > L. monocytogenes
Line 427 Pattern-Recognition Receptor > Pattern-Recognition Receptors
Line 428 “Pattern-recognition receptors are believed to be expressed and may produce… > I’m not sure if this sentence transports the intended meaning.
Line 468: In the study of S. suis > In a study of S. suis
Re:
Thanks for the careful review on our manuscript. We have corrected these above mistakes in our revised manuscript, please check the latest version.
Round 2
Reviewer 2 Report
I appreciate the efforts of authors to enrich the manuscript.
Please consider the following minor comment
Line 480: modify "mortality and morbidity of" to "mortality and morbidity due to"
Line 481: modify "remain" to remains
Line 491: Modify "approach to" to "approach in the"
Author Response
I appreciate the efforts of authors to enrich the manuscript. Please consider the following minor comment
Line 480: modify "mortality and morbidity of" to "mortality and morbidity due to"
Re:
Thanks for your correction. Please check the revised manuscript Line 480.
Line 481: modify "remain" to remains
Re:
Thanks for your correction. Please check the revised manuscript Line 481.
Line 491: Modify "approach to" to "approach in the"
Re:
Thanks for your correction. Please check the revised manuscript Line 491.
Reviewer 6 Report
In the revised version of their manuscript the authors have answered to the comments and significantly enhanced the quality of the manuscript. I believe that the manuscript is now suited for publication in the International Journal of Molecular Sciences.
Typos / corrections:
Line 42: such cerebral palsy > such as cerebral palsy
Line 169: Gram-Positive > Gram-positive
Line 479: by using antibiotic > by using antibiotics
Author Response
In the revised version of their manuscript the authors have answered to the comments and significantly enhanced the quality of the manuscript. I believe that the manuscript is now suited for publication in the International Journal of Molecular Sciences.
Typos / corrections:
Line 42: such cerebral palsy > such as cerebral palsy
Re:
Thanks for your correction. Please check the revised manuscript Line 42.
Line 169: Gram-Positive > Gram-positive
Re:
Thanks for your correction. Please check the revised manuscript Line 169.
Line 479: by using antibiotic > by using antibiotics
Re:
Thanks for your correction. Please check the revised manuscript Line 479.